# Molecular Approach for the Laboratory Diagnosis of Periprosthetic Joint Infections

**DOI:** 10.3390/microorganisms10081573

**Published:** 2022-08-05

**Authors:** Giulia Gatti, Francesca Taddei, Martina Brandolini, Andrea Mancini, Agnese Denicolò, Francesco Congestrì, Martina Manera, Valentina Arfilli, Arianna Battisti, Silvia Zannoli, Maria Michela Marino, Anna Marzucco, Manuela Morotti, Laura Grumiro, Agata Scalcione, Giorgio Dirani, Monica Cricca, Vittorio Sambri

**Affiliations:** 1Unit of Microbiology, The Great Romagna Hub Laboratory, 47522 Pievesestina, Italy; 2Department of Experimental, Diagnostic and Specialty Medicine—DIMES, Alma Mater Studiorum—University of Bologna, 40138 Bologna, Italy

**Keywords:** prosthetic joint infection, PJI, molecular diagnosis, PCR, NGS

## Abstract

The incidence of total joint arthroplasty is increasing over time since the last decade and expected to be more than 4 million by 2030. As a consequence, the detection of infections associated with surgical interventions is increasing and prosthetic joint infections are representing both a clinically and economically challenging problem. Many pathogens, from bacteria to fungi, elicit the immune system response and produce a polymeric matrix, the biofilm, that serves as their protection. In the last years, the implementation of diagnostic methodologies reduced the error rate and the turn-around time: polymerase chain reaction, targeted or broad-spectrum, and next-generation sequencing have been introduced and they represent a robust approach nowadays that frees laboratories from the unique approach based on culture-based techniques.

## 1. Introduction

Total joint arthroplasty (TJA) has improved the quality of life of surgery patients in the past half-century by relieving them from pain and restoring a high level of mobility [1]; according to some outlooks, the number of surgical interventions will rise over the coming years, essentially due to the increased aging of the population [2]. More than 4 million joint replacements will be annually expected by 2030 and, in this scenario, periprosthetic joint infections (PJIs) represent some of the most challenging complications whose incidence will consequently rise in proportion to surgeries [3,4]. The manifestation and evaluation of physical findings such as acute local inflammation, fever, and wound drainage, may correlate to the presence of PJIs; indeed, these clinical manifestation are of great value in raising the suspicion of PJIs. Nevertheless, the suspected diagnoses have to be validated by an examination of biological samples obtained through aspiration or by biopsy, with the purpose of identifying the etiological agent and defining the proper pharmacological and surgical treatment strategy; in some cases, one-stage exchange revision surgery could be relevant in the identification of the infectious agent [4]. However, PJIs occur less frequently than aseptic failures, dislocation, and bone or periprosthetic fractures, but infections represent the most severe collateral damage. Microbiological assessment by the evaluation of the patient’s history and exams is playing an essential role in the discrimination between PJIs and different causes, such as an aseptic loosening. The standard methods for the microbiological diagnosis of PJIs relies on the inspection and in vitro culture of tissue specimens surrounding the prosthesis, differential of synovial fluid obtained by arthrocentesis, and culture of intraoperative collected specimens [5].

## 2. Epidemiology of Periprosthetic Joint Infections

The practice of arthroplasty is mainly directed towards the elderly population; therefore, the number of surgeries is continuously rising with the increase in the life expectancy. Although, in numerous surgical interventions, less than 10% of recipients develop complications and among these, less than 1–3% are microbial infections of primary arthroplasties [6,7]. PJIs can be related to a high morbidity rate and are responsible for severe transient or permanent disabilities, as for instance, arthrodesis or leg amputation, but also lead to complicated socioeconomic implications [6], so much that the cost of the management of PJI affected patients subject to knee and hip TJA has been evaluated as being over a billion USD in 2017 and USD 1.85 billion by 2030 [8]. The incidence has been reduced by the introduction of laminar airflow systems in the routinary perioperative antimicrobial prophylaxis of operating rooms [6].

The rate of infection within the first two postoperative years after primary arthroplasty ranges from <9% in elbow prostheses, <4% in knee prostheses, and <2% in hip and shoulder prostheses [6,7]. In this context, the prevalence of hip and knee TJA is expected to double by year 2030 [8].

In contrast, the infectious incidence rises up to 40% after revision surgery: it has been estimated that PJIs are the cause of 15% of all revision hips surgery and 25% of all revision knee TJA. The mortality rate is 5 years higher than melanoma, Hodgkin’s lymphoma, and breast cancer [6,8].

Moreover, the importance of management and treatment of PJI represents a crucial issue for the lifetime condition of patients with inflammatory joint diseases such as rheumatoid arthritis, juvenile inflammatory arthritis, ankylosing spondylitis, and psoriatic arthritis who are more susceptible to infection after arthroplasty [1,2,6].

The diagnoses of infections after arthroplasty may be underestimated since many cases of aseptic failure are actually caused by low-virulent bacteria and can require costly and intensive interventions for an effective treatment [6,8].

## 3. Periprosthetic Joint Infections Definition

The achievement of an accurate definition of PJI has gathered in 2011 by a workgroup of specialists for the Musculoskeletal Infection Society that indicated two major criteria and five minor criteria for the definition of infections following a prosthesis implantation [1,9]. In order to diagnose a PJI, the physical inspection has to fulfil one of the two major criteria and at least three minor characteristics [9]. A synthetic table of PJI criteria can be found in the review of Premkumar A. et al. [1].

Later, in an International Consensus Meeting in 2014 and 2018, the fundamental guidelines for the diagnosis of PJI have been introduced [10].

Lastly, in 2021, the European Bone and Joint Infection Society (EBJIS) committee has proposed a new definition of PJI based on three levels of evidence [11].

An important progress for the diagnosis of an infection associated with an arthroplasty has been provided by the introduction of new molecular diagnostic techniques [4,12].

## 4. Periprosthetic Joint Infections Classification

The period of symptom manifestation can vary and depends on microbial virulence [6,10]. Infections that have an outbreak within the first month after implantation are called early infections: high-virulent causative agents as *Staphylococcus aureus*, *Streptococci* spp., and *Enterococci* spp. elicit an evident local and systemic inflammation. On the contrary, low-virulent organisms such as *Cutibacterium* spp. and coagulase-negative *Staphylococci* (CoNS) might be the starting agents for infections manifesting within three months and one year after the implantation of the prostheses. These types of infection are named as delayed, and they show mainly attenuated symptoms such as joint pain or early loosening. Late infections present more than 24 months after surgery [10,13]. Not only may the bacterial colonization of the periprosthetic tissue represent a beginning factor for an inflammatory response, but also the high vascularity of those areas predisposes the patient to the risk of hematogenous infections within the first year after surgery [6,10]. Hence, the detection and elimination of the originating seeding focus is mandatory to prevent a relapse. The most common infectious bacterial species for hematogenous spreading are:*Staphylococcus aureus*, from skin and soft tissues infections that possibly leads to bacteraemia up to 34%;*Streptococcus pneumoniae*, spreading from respiratory tract;*Salmonella*, *Bacteroides*, *Streptococcus gallolyticus* from gastrointestinal infections;*Escherichia coli*, *Klebsiella*, *Enterobacter* spp. Affecting the urinary tract.

Other low-virulent bacteria such as *Staphylococcus epidermidis* may be responsible for a systematic infection, likewise the viridans group, *Streptococci*, can spread in the general blood circulation starting from dental procedure infections [10].

Notably, patients who underwent second-stage reimplantation and a subsequent doxycycline treatment can be colonized by *S. epidermidis* resistant to the molecule. This event appears be critical because doxycycline is a well-tolerated long-term use antibiotic [14].

Infections related to shoulder prostheses may relate to *Cutibacterium acnes* (formerly *Propionibacterium acnes*) [15,16], a low-virulent anaerobic Gram-positive bacterium, and a commensal skin microbe that exists in different subtyping populations [2,17]. An abnormal distribution could lead to a failure and reversion surgery [17].

*Staphylococcus lugdunensis* is CoNS that generally colonizes hips and perineum and mutates under stress conditions as a limitation of nutrients or sub-lethal concentrations of antibiotics, inducing the formation of genetically modified small-colony variants (SCVs). Regarding pathogenicity, *S. lugdunensis* resembles *S. aureus* and might be misidentified in slide coagulase tests rather than tube tests because of the production of yellow pigments and DNase [18].

Infections caused by a direct contact are called *per continuitatem* and subdivide into two classes: a direct contact between the prostheses and the external world or a spread from a nearby infectious focus (e.g., osteomyelitis) [10].

Additionally, fungal PJIs are under-reported consequences in 1–3% of a failed joint arthroplasty and the symptomatologic profile is complicated in the presence of comorbidities and susceptibility to antifungal drugs. Vast majority of diagnoses due to a single infectious agent are attributed to *Candida* spp. and in particular to *C. albicans*. *Candida* species adapt to rapidly changing environments and express an adhesion capability to bind native tissues or implanted devices. Analogously to *C. albicans*, *Aspergillus* spp. present a dimorphic form between the growing hyphal structure and unicellular yeast or spores state, that is more difficult to eradicate [19].

Despite clinical evidence as purulence, sinus tract or histological acute inflammation, some hypotheses of infection, 7% to 15%, may result in culture-negative response whose main cause is a prior antimicrobial therapy of the patient [2].

## 5. Periprosthetic Joint Infections Biofilm

One of the main bacterial defensive strategies implicates the self-production of a polymeric matrix that provides protection when the environmental conditions are hostile for one or multiple species. The biofilm takes four weeks to develop [10,20], encloses the colony to form a dense bacterial plaque where nutrients can circulate and adheres to inert or living surfaces, as the case of bacterial endocarditis; likewise, medical devices and prostheses represent a possible colonizing area. The presence of the biofilm indices and the production of antibodies that are not able to eradicate the infection since the penetration results are problematic; therefore, the immune response may induce a damage to the surrounding tissues due to the synthesis of immune complexes. Accordingly, the penetration of substances depends on the composition of the polymeric matrix [20,21] that confers tolerance and makes the infection difficult to diagnose and eradicate [6,21,22], as difficult-to-treat PJI caused by *staphylococci*, Gram-negative bacteria, *Enterococci*, and *Candida* spp. are resistant to anti-biofilm agents such as rifampicin [23].

Several bacterial species can form a biofilm plaque on orthopaedic implants, including *S. aureus*, *S. epidermidis*, *Enterobacter* spp., *Enterococcus faecium*, *Klebsiella pneumonie*, *Acinetobacter baumannii*, and *Pseudomonas aeruginosa* that are typically found in PJI examination [2,22]. Coagulase-negative *Staphylococci*, and especially *S. aureus*, are the major pathological causes of PJI, that together with *Staphyloccocus* epidermidis and *Pseudomonas aeruginosa*, represent the 75% of causative biofilm forming pathogens for PJI: in particular, *S. aureus* infections triggers 20–25% of infections [2,22,24]. Numerous studies have addressed the attention to the decolonization of operation rooms from *S. aureus* prior a surgical intervention. However, the effectiveness of the protocol lacks international guidelines by international societies [25].

A novel antiplatelet molecule, Ticagrelor (a P2Y12 receptor inhibitor), has been studied in *S. aureus* related to infection with biofilm formation, both in vitro and in vivo. Ticagrelor has shown a synergic effect with rifampicin, ciprofloxacin, and vancomycin. A therapy based on Ticagrelor may improve the rate of PJI treatment [26].

*S. aureus* biofilm elicits an anti-inflammatory response mediated by the recruitment of myeloid-derived suppressor cells, leading to the activation of monocyte and macrophages via IL-10. Additionally, the signal pathway of Toll-like receptors inhibits the macrophage phagocytosis and favours the persistence of the bacterium on the surface of the device [27].

Species as *Candida* form biofilm on the surface of medical devices and potentially disseminate within the systematic circulation causing hematogenous infections [28].

*Candida* demonstrates a synergic relationship with *S. aureus* that increase the progression and growth rate of the infection. On the contrary, a massive use of broad-spectrum antibiotics increases the probability of fungal infections, remarkably, due to *Aspergillus* spp. [20].

The mere presence of colonizing bacteria is necessary for the formation of a biofilm, and is assisted by certain common risk factors as immunomodulation or steroidal therapy, renal diseases, diabetes, smoking, an imbalanced body mass index, and age. In the general population, the mean percentage to develop a biofilm PJI in a model distribution which considers a uniform risk distribution is 1%. Differently, the population can be subdivided into two classes, high-risk (representing the 10% of cases) and low-risk (the remaining 90%) biofilm infection development cases. Patients in the high-risk group have a 10% risk to forming biofilm on a device surface, whereas the 90% low-risk population approximates its average percentage to zero [29].

The current therapies for the eradication of the biofilm are sufficiently functional; therefore, the prevention of implant colonization either preventing the attachment (antifouling) or killing bacteria by surface contact (bactericidal surfaces) may represent a valid prevention [30].

## 6. Periprosthetic Joint Infections Diagnosis

Since the annual increase in primary and reversion arthroplasties, the need for a rapid and accurate diagnosis is ever more crucial for the management of PJI and an early bacteria detection, particularly within the first 2 weeks, allows a timely treatment of the biofilm [31].

The diagnosis is based on a combination of clinical findings and laboratory results obtained from multiple tests conducted through different techniques as peripheral blood, histological tissues, and microbiological cultures evaluation and intraoperative findings evaluating numerous markers as D-dimer, C-reactive protein, synovial leukocyte esterase, and synovial alpha-defensin [32].

Subsequent to clinical inspections, the following step is the culture of synovial fluid, and the evaluation of chemical and biochemical parameters through Gram coloration, especially for Gram-positive *Staphylococcus aureus*, *Streptococcus* spp., and Gram-negative diplococci [33,34]. Regardless many methodologies as sonication [24,35], or detection of serum inflammatory markers (C-reactive protein, erythrocyte sedimentation rate synovial fluid cell count) [36]. In regard to the analysis of sonication fluid cultures, 16S r RNA beacon-based fluorescent in situ hybridization (bbFISH) represents a novel molecular approach for the identification of bacterial pathogen in PJI [37].

Typically, the gold standard for the diagnosis of PJI remains a tissue culture examination [34]. The definition of a PJI diagnosis is made on the positivity of two or more periprosthetic tissue cultures for the same microorganism. However, conventional cultures may be error-prone because of an appropriate medium, short incubation time, loss of microbial load due to conservation conditions, or a prior antimicrobial therapy [5].

Imaging techniques are also used into the diagnose of PJI, although their low sensitivity and specificity. Computed tomography (CT) or magnetic resonance imaging (MRI) that provide a good resolution for soft tissues [38].

One of the major challenges that remains is the culture-negative PJI caused by numerous reasons as antibiotics treatment, low-virulent bacterial infections, or biofilm. In patients who are not capable to undergo surgery, a proper antibiotic treatment has a success in 23–83% of cases; however, the lack of standardization and the vast surgical and antibiotics options make the resolution difficult to achieve [39,40].

In the last years, the implementation of molecular biology techniques has reduced the turn-around time to few hours and completely automated the workflow [4]: specific real-time, broad range 16S-polymerase chain reaction (PCR) [5], or next-generation sequencing that could be a robust support for PJI diagnosis [31]. The application of an interdisciplinary approach of surgery, infectiology, and microbiology will reduce the incidence of PJI and prolong the infection-free survival time of patients [41].

## 7. Periprosthetic Joint Infections Molecular Diagnosis

The PCR represents a robust diagnostic tool for PJI that can be applied to different specimens: tissue samples, synovial fluid, or sonicated prosthetic fluid [42].

The sensitivity and the velocity of the PCR overtakes the ones of tissue cultures; therefore, it is applied into the characterization of the bacterial colonies [43].

The PCR target can be amplified for a specific single organism or for multiple bacteria, acquiring the name of multiplex-PCR in this instance [44].

The nature of a PCR technology can be a house-made methodology or a commercial kit; the establishment of a robust protocol remains a challenging perspective because of the standardization: the reproducibility often requires well-trained technicians. For that purpose, many pharma companies have commercialized multiplex-PCR protocols: examples are SeptiFast by Roche, Genotype by Hain, Xpert by Cephaid, Filmarray by Biofire [45]. Culture-negative infections can be detected and diagnosed through multiplex-PCR, also defining the gene profile of microorganisms through the comparison to databases [46].

On the other hand, a broad range PCR targeting the 16S gene has a sensitivity from 50% to 92% and a specificity from 65% to 94%. Broad range PCR remains susceptible to error; hence, the lack of sensitivity may not correctly identify the microorganism leading to a false-positive result; on the contrary, all bacteria can be identified in a polymicrobial infection reducing the turn-around time [44,46,47].

Lastly, the detection of *Kingella kingae*, an important paediatric osteoarticular infectious agent, requires a pathogen-specific PCR on 16S gene [44].

The false-positive yield of broad-range PCR remains an issue of concern: in order to reduce the misdiagnosis and contamination, some studies tried to develop genus-specific PCR targeting a subgroup of Gram-positive cocci and excluding *E.coli* [48].

In case of mixed cultures, the broad-range PCR can detect the dominant bacterial strain present in the infection site; hence, the definition of the proper causative microorganism is appointed to NGS. Through PCR, non-viable microorganisms can be identified in a small volume and within few hours [41], also in cultures of patients receiving an antibiotic treatment. Particularly, PCR can implement the culture-independent diagnosis [44].

A limitation of the PCR technique could be contamination: a proper quality control process could prevent a false-positive result [48,49].

The incapacity to distinguish between living or dead bacteria and DNA contamination represent the main limitations of PCR-based diagnosis yielding to a false positive result [50].

In 2021, Bourbon et al. have associated the high-resolution melt analysis (HRMA) to broad-range PCR positive specimens. Equally to PCR, HMRA targets 16S gene and relates to the denaturation of the bacterial DNA and the melting temperature depending on the G-C content of the sequence and its length. The detachment of the double strand DNA decreases the fluorescent emission intensity that represents a rapid and cost-effective method for bacterial detection [51].

A broad-range PCR is followed by Sanger or next-generation sequencing (NGS) and exploits the high conservation of the 16S region that varies in short subunits depending on the bacteria species [44].

## 8. Periprosthetic Joint Infections Sequencing

In the last years, the NGS has acquired an increasing important role in PJI diagnosing. This technique completely frees laboratory procedures from a culture-based approach and raises the sensitivity of protocols; the enhanced sensitivity is counterbalanced by a decrease in specificity leading to probable false positive results [52].

Culture-negative infections constitute about 7–12% of all PJI whose 16–44% can be diagnosed with a new potential pathogenic microorganism through metagenomic NGS. In 4–67% of culture-positive infections, a novel pathogen can be equally discovered by a metagenomic analysis [53].

The potential of whole genome sequencing also provides an accurate characterization and reconstruction of the infection chain and outbreak together with a rapid identification of antimicrobial resistance to antibiotics by querying a database or a reference sequence [54].

Equal to broad-range 16S PCR, NGS allows for the identification of a large range of pathogens; through sequencing, even fungal infections are diagnosed implying primers for internal transcribed spacer (ITS) or 18S rRNA gene. However, when compared to broad-range PCR, NGS enables the detection of a crowded field of pathogens [55,56].

Morales-Laverde et al. have implied the Illumina whole genome NGS on *S. aureus* spp. isolated from a PJI site for the identification of single nucleotide polymorphisms (SNPs). Mutations were investigated in the regulatory regions (IGR) for genes encoding for biofilm matrix compounds as β-1,6- linked N-acetylglucosamine [57].

Illumina whole genome sequencing was implied in the study of Wildeman and al. for *S. aureus* resistant to rifampicin and fluoroquinolone which are active against microbial biofilm [58].

An additional study on *S. aureus* periprosthetic infections was the one conducted by Sanabria et al. by shotgun-metagenomics (SMg) analysis. The resistances to penicillin and fusidic acid were indeed investigated, resulting in the prevalence of 63.5% and 10.5%, respectively [59].

In the study of Goswami et al., three NGS techniques were compared: 16S rRNA amplicon sequencing, and untargeted shotgun metagenomics and metatranscriptomics (first application of RNA-based sequencing) for the identification of PJI causative pathogen. Particularly, the investigation of gene expression can be through the metatranscriptomics; therefore, pathogenic pathways, functional antimicrobial-resistance, and active virulence mechanisms can be disclosed [60].

## 9. Conclusions

The number of joint replacements will increase annually by 2030: PJI represent the most challenging complications to these procedures [3] and *S. aureus* appears as the major causative pathogen [10], also responsible for biofilm-mediated inflammation [27].

PCR serves as a robust diagnostic tool for PJI and can be applied on different biological matrixes [42].

In recent years, the NGS has acquired a relevant diagnosing relevance for PJI by freeing laboratories from culture-based approaches and increasing sensitivity [52].

Indeed, culture-negative infections can be properly detected and diagnosed through NGS [53].

A significant step forward for the diagnosis of PJI could be represented by the integration of different diagnostic approaches directed at the formation of a multidisciplinary team and the implementation of culture-based techniques using molecular biology [44].

Through PCR, complications related to culture-negative results can be solved through the increased sensitivity of the method, although the limitation of the technique remains the use of specific primers depending on the investigated organism [61,62]. Similar to PCR, NGS could provide a rapid and precise response [61].

Difficulties in the application of molecular techniques have been reported in some laboratories [63]. In particular, issues related to the use of NGS in the diagnostic procedure of PJIs are caused by the lack of amplification, hence sequencing, due to the length of primers implied. On the contrary, an extreme sensitivity could lead to a false positive result [64]. NGS can reduce the turn-around time, be performed directly on the native material, and generate a significant amount of data utilized also in an in-depth detection of antibiotic microbial resistances. With computational methods, the background noise in the analysis is removed and the precision of the analysis is increased [65].

Despite some limitations, techniques such as PCR or NGS can strengthen the diagnosis of PJI when applied with proper consideration. Molecular biology approaches find utility in different fields and can implement gold standard methods.

## Data Availability

Not applicable.

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
