# Peer review of "Molecular Approach for the Laboratory Diagnosis of Periprosthetic Joint Infections"

_microorganisms, 2022, doi:10.3390/microorganisms10081573_

Round 1
Reviewer 1 Report
This is a very well writen review. Good for publication.
Minor changes required:
1. Line 126. DNAse to DNase
2. Line 154 staphylococci should be italicized
3. Line 175 Aspergillus spp. should be italicized
4. Line 282 spp. should be italicized
Author Response
We are sincerely thankful to Reviewer 1 for their evaluation and corrections. The following modifications were made at lines:
- 132:”DNAse” was modified in “DNase”
- 160: “staphylococci” italicized
- 185: “Aspergillus spp.” Italicized
- 303: “ssp.” Italicized
Reviewer 2 Report
Dear authors,
First, I would like to express sincere gratitude to get an opportunity to review your manuscript.
The effort of the author is appreciated. It was my pleasure to assess your manuscript. Congratulations on the selection of the subject of your literature review. After assessing the manuscript, the following issues raised my concerns or represent suggestions that in my opinion could increase the quality of the manuscript:
- Introduction
o “immediate revision surgery” – if it is possible please use another them, Debridement, antibiotics, and implant retention (DAIR) or One-stage exchange or Two-stage exchange depending on what type of surgery you had in mind.
o “ prosthesis fracture” – please change with periprosthetic fractures
- PJI definition
o You mentioned the MSIS 2011, ICM 2013, and 2018, I would suggest also mentioning the EBJIS criteria that were published on 2021 1st Jan.
- PJI classification
o “Propionibacterium acnes (or Cutibacterium acnes)” – please change to Cutibacterium acnes (formerly Propionibacterium acnes) ("Genus: Cutibacterium". Prokaryotic Nomenclature Up-to-Date. DSMZ. Archived from the original on 17 August 2018. Retrieved 17 August 2018. Scholz CF, Kilian M (November 2016). "The natural history of cutaneous propionibacteria, and reclassification of selected species within the genus Propionibacterium to the proposed novel genera Acidipropionibacterium gen. nov., Cutibacterium gen. nov. and Pseudopropionibacterium gen. nov" (PDF). International Journal of Systematic and Evolutionary Microbiology. 66 (11): 4422-4432. doi:10.1099/ijsem.0.001367. PMID 27488827. Retrieved 17 August 2018. Pcastaings S, Corvec S, Veraldi S, Khammari A, Roques C (June 2018). "Cutibacterium acnes (Propionibacterium acnes) and acne vulgaris: a brief look at the latest updates". Journal of the European Academy of Dermatology and Venereology. 32 Suppl 2: 5-14. doi:10.1111/jdv.15043. hdl:2434/620522. PMID 29894579. Genus Cutibacterium”. LPSN. Retrieved 17 August 2018.
- PJI diagnosis
o “the gold standard for the diagnosis of PJI remains a tissue culture examination [25].” What about sonication fluid cultures? There is also data regarding the use of molecular techniques on the sonication fluid like bbFISH.
The NGS and PCR-based techniques definitely increase the diagnosis capabilities. PCR-based techniques are frequently used in the field of PJIs. The use of NGS in routine clinic not in the research field at the moment in my opinion is not feasible. We should also pay attention at the costs and the disadvantages/errors.
Author Response
We express our sincere gratitude to Reviewer 2 for corrections and their contribution to the manuscript. The following modifications were made at lines:
- 43: “one-stage exchange” was specified
- 45: “prosthesis fracture” was modified in “periprosthetic fractures”
- 88-89: “Lastly, in 2021, the European Bone and Joint Infection Society (EBJIS) committee has proposed a new definition of PJI based on a three-levels evidence [11].”
Reference:- 11 - McNally M, Sousa R, Wouthuyzen-Bakker M, et al. The EBJIS definition of periprosthetic joint infection. Bone Joint J. 2021;103-B(1):18-25. doi:10.1302/0301-620X.103B1.BJJ-2020-1381.R1
- 122-123: “Propionibacterium acnes (or Cutibacterium acnes)” was modified in “ Cutibacterium acnes(formerly Propionibacterium acnes) [15,16]”
References:- 15 - Scholz CFP, Kilian M. The natural history of cutaneous propionibacteria, and reclassification of selected species within the genus Propionibacterium to the proposed novel genera Acidipropionibacterium gen. nov., Cutibacterium gen. nov. and Pseudopropionibacterium gen. nov. Int J Syst Evol Microbiol. 2016;66(11):4422-4432. doi:10.1099/ijsem.0.001367
- 16 - Dréno B, Pécastaings S, Corvec S, Veraldi S, Khammari A, Roques C. Cutibacterium acnes (Propionibacterium acnes) and acne vulgaris: a brief look at the latest updates. J Eur Acad Dermatol Venereol. 2018;32 Suppl 2:5-14. doi:10.1111/jdv.15043
- 216-217-218: “In regards to the analysis of sonication fluid cultures, 16S r RNA beacon-based fluo-rescent in situ hybridization (bbFISH) represents a novel molecular approach for the identification of bacterial pathogen in PJI [37].” was added
Reference:
- 37 - Birlutiu RM, Birlutiu V, Cismasiu RS, Mihalache M. bbFISH-ing in the sonication fluid. Medicine (Baltimore). 2019;98(29):e16501. doi:10.1097/MD.0000000000016501
Round 2
Reviewer 2 Report
Dear authors,
Again, I would like to express sincere gratitude to get an opportunity to review your manuscript. You greatly improved the quality of the manuscript with this revision. A manuscript that in my opinion deserves to be published.